# Coping as a Mediator between Attachment and Depressive Symptomatology Either in Pregnancy or in the Early Postpartum Period: A Structural Equation Modelling Approach

**DOI:** 10.3390/brainsci13071002

**Published:** 2023-06-28

**Authors:** Mario Altamura, Ivana Leccisotti, Laura De Masi, Fiammetta Gallone, Livia Ficarella, Melania Severo, Simona Biancofiore, Francesca Denitto, Antonio Ventriglio, Annamaria Petito, Giuseppe Maruotti, Luigi Nappi, Antonello Bellomo

**Affiliations:** 1Department of Clinical and Experimental Medicine, University of Foggia, 71122 Foggia, Italy; ivana.leccisotti@unifg.it (I.L.); laura.demasi@unifg.it (L.D.M.); fiammetta.gallone@unifg.it (F.G.); livia.ficarella@unifg.it (L.F.); melania.severo93@gmail.com (M.S.); simona.biancofiore@gmail.com (S.B.); francesca.denitto@unifg.it (F.D.); antonio.ventriglio@unifg.it (A.V.); annamaria.petito@unifg.it (A.P.); antonello.bellomo@unifg.it (A.B.); 2Department of Medical and Surgical Sciences, University of Foggia, 71122 Foggia, Italy; maruotti.g@gmail.com (G.M.); luigi.nappi@unifg.it (L.N.)

**Keywords:** perinatal depression, attachment style, coping strategies, gender medicine, structural equation modelling

## Abstract

Peripartum depression (PPD) is a major complication of pregnancy, and numerous risk factors have been associated with its onset, including dysfunctional coping strategies and insecure attachment styles, both during pregnancy and postpartum. The aim of our study was to investigate the role of coping strategies in mediating the relationship between women’s attachment style and depressive symptomatology in pregnancy and one week after giving birth in a large sample of women (N = 1664). Our hypothesis was that the relationship between anxious and avoidant attachment and depressive symptomatology would be mediated by use of maladaptive coping strategies. The assessment instruments were Edinburgh Postnatal Depression Scale (EPDS), Brief Coping Orientation for Problem Experiences (COPE), and Experiences in Close Relationship Scale (ECR). The results indicated that the effect of insecure attachment styles (anxious and avoidant attachment) on antepartum depressive symptomatology was partially mediated by dysfunctional coping styles. Anxious attachment also has an indirect significant effect on postpartum depressive symptomatology through emotional coping; however, avoidant attachment does not seem to be significantly related to postpartum depressive symptoms. Our findings revealed that not only is it important to consider attachment in understanding peripartum depressive symptomatology, but also that coping plays an important role in these relationships. These findings would help a preventive coping-based intervention strategy to enhance the capacity of women with insecure attachment styles to use more adaptive coping during and after pregnancy.

## 1. Introduction

Major depressive disorder with peripartum onset is the most common complication of childbearing. Estimates show that one in seven women can experience peripartum depression (PPD) occurring during pregnancy or after childbirth [1,2]. A wide range of factors have been consistently identified as important risk factors that interact to contribute to the development of depressive symptomatology, including genetic predisposition, and various environmental, social, and psychological factors including negative personality traits (e.g., neuroticism, insecure attachment styles) and ineffective coping strategies (e.g., denial, substance use) [3,4,5].

Coping is described as the cognitive and behavioral strategies that people may use to deal with stressful circumstances that are demanding, challenging, threatening, and/or have a potential for harm or loss [6]. Growing evidence suggests that maladaptive coping strategies (e.g., emotional coping, denial) may be risk factors for depressive mood during both antenatal [7,8] and postnatal period; while adaptive coping strategies (e.g., acceptance, humor) tend to be protective factors for PPD [3,8,9,10]. For instance, a recent large prospective study of 1626 women reported that maladaptive coping strategies such as substance use, self-distraction, and self-blame were positively associated with major postpartum depression [11]. Contrastingly, women who have positive coping styles that involve actively solving problems and positively interpreting situations were less likely to have perinatal depressive symptoms [12,13,14,15]. In recent years, research has also shown that coping with stressful conditions to ensure proper psychological adjustment does not operate in isolation. Rather, coping strategies appeared to mediate the relationship between personality traits and depressive symptomatology [11,12,16].

Preparation and transition to motherhood is considered a stressful experience that demands several adjustments to a new role and new responsibilities, which clearly triggers the attachment system throughout pregnancy and beyond [17]. The attachment system is an innate behavioral system that is biologically driven to promote survival through relational closeness. According to attachment theory, the infant’s early experiences with caregivers determine internal working models of attachment (a set of mental representations of the self and others) which influence the individual’s future attachment behaviors in intimate relationships, including adult romantic relationships and parenting [18,19,20,21]. When a caregiver is seen as being trustworthy and accessible, an infant establishes a secure attachment style. In contrast, in situations where the primary caregiver is perceived as being unavailable and/or unresponsive, an infant develops a negative internal working model and an insecure style of attachment [22]. Researchers have identified two types of insecure attachments in adults: anxious attachment and avoidance attachment. The nature of anxious attachment is distinguished by an overly active attachment mechanism that results in an ongoing quest for reassurance and consolation. On the other hand, avoidant attachment is defined by a less active attachment mechanism, leading to an ongoing suppression of needs related to psychological and social connections. This also promotes an excessive reliance on oneself and cultivates a negative disposition towards others [23]. Several studies have shown that insecure attachment styles are associated with depressive symptoms in adulthood [24,25,26]. Particularly, many previous studies have consistently shown that insecure attachment styles are strong and independent predictors of depression symptomatology either during pregnancy or in the period following birth (see for review [4,27]). For instance, Bianciardi et al. [5] found that insecure attachment styles were associated with depressive symptoms during both antenatal and early postpartum phases. However, many scholars have questioned the direct path between attachment and depression, and instead, they argue that early attachment experiences do influence later psychological distress through different intermediate mechanisms such as coping strategies [28,29,30]. Over the past few decades, research has provided extensive evidence suggesting that the coping strategies that individuals employ in stressful situations are determined, at least in part, by their attachment style [29,31]. Securely attached persons tend to rely on active coping styles (e.g., support-seeking coping strategies) and to maintain acceptable psychological well-being during stressful periods (e.g., [32,33,34]).

In contrast, individuals with avoidance or anxiety attachment styles have been found to appraise stressful events in threatening terms and report doubts about their coping abilities [35]. For instance, Schmidt et al. [36] found insecure attachment to be related to less flexible coping, with anxiously attached individuals showing more negative emotional coping (e.g., impulsivity and distortion) and avoidantly attached individuals showing more disengaging methods of coping (e.g., denial, tendency to avoid awareness of problems). 

Despite many previous studies reporting a significant relationship between attachment styles and coping strategies, a mediational model to explore how coping might mediate the link between attachment and perinatal depressive symptomatology has not been performed. Insights from attachment theory and behavioral neuroscience are not just theoretically interesting, but they have significant implications when applied to patients dealing with depressive disorders that occur during the perinatal period. Recognizing potential factors like coping mechanisms, which mediate the connection between attachment and depression, could enhance treatment results. This can be achieved by choosing therapies based on attachment that target either insecure attachment styles or the maladaptive coping methods linked to these styles. 

Coping styles were originally divided into two high-order dimensions: problem-focused coping styles and emotion-focused coping styles [37]. However, there are many ways to group coping responses. An important distinction is between problem-focused (e.g., active coping, planning), emotion-focused coping (e.g., support seeking, emotion regulation), and disengagement coping, which includes maladaptive coping strategies such as avoidance, denial, and wishful thinking [38,39,40,41]. However, in the existing literature, uniformity is lacking in terms of how coping strategies are grouped to form higher-order factors. More research is necessary to ascertain the suitable factor structure, contingent upon the context [42]. Considering that higher-order strategies, including emotion-focused coping and maladaptive coping, have been found to consist of other lower-order coping strategies, it would be preferable to conduct individual factor analyses and determine the factor structure in the context of peripartum depressive symptomatology, instead of relying on previous factors that have already been defined [29]. Thus, in the present study, two steps are envisaged. First, we use exploratory and confirmatory factor analysis (CFA) to investigate the factor structure of the coping. Second, we examine the role of coping strategies in mediating the relationships between insecure attachment styles and depressive symptomatology in both the antepartum and postpartum periods. In this regard, structural equation modelling (SEM) analysis is a suitable statistical method to evaluate the extent to which a third intermediate or mediating variable (coping) explains the effect of a predictor (e.g., attachment insecurity) on an outcome (e.g., depressive symptoms) [43].

Depression is significant at each stage but most reported in the second half of pregnancy and early postnatal period [44,45] The present study aimed to examine the relationship of coping strategies, attachment styles, and symptoms of depression that occur in the late pregnancy period (third trimester) and within early postpartum period (day 1–7) as previously reported [45]. Based on the literature revised, we hypothesized that (a) both anxious and avoidant attachment scores, respectively, would be significantly associated with dysfunctional coping strategies; and (b) the relationship between anxious and avoidant attachment and perinatal depressive symptomatology would be mediated by the use of dysfunctional maladaptive coping strategies.

## 2. Materials and Methods

The data of this study were collected as part of a previous longitudinal study that assessed the prevalence and risk factors of perinatal depression among pregnant women attending, from July to November 2020, maternal health clinics in three major public hospitals (Ospedali Riuniti di Foggia, Ospedale Vito Fazi di Lecce, and Ospedale Di Venere di Bari) of the Regione Puglia (Italy) [46]. The inclusion criteria were being a woman over the age of 18 and in the third trimester of pregnancy. The exclusion criteria included intellectual disability and poor knowledge of language which undermined the possibility of the woman participating in the research protocol. All the participants were given verbal and written explanations of the purpose of the study, and their informed consent was obtained prior to their participation in the study. During the study period, 1664 eligible women were enrolled and assessed during the third trimester of pregnancy (t0). The same cohort was followed up one week after delivery (t1) [47,48]. Sociodemographic characteristics, attachment styles, and coping strategies were assessed during the first visit (t0). The characteristics of patients with and without depression were subjected to an analysis using descriptive analysis. T-test was applied for continuous data. 

### 2.1. Edinburg Postnatal Depression Scale

The validated Italian version of the Edinburgh Postnatal Depression Scale (EPDS) [49,50] was administered twice, once during the antenatal visit and again during a postnatal follow-up. The EPDS was initially configured by Cox et al. [51] to detect the level of maternal depressive symptoms present in the first weeks after childbirth and, consequently, to identify women at psychopathological risk through a clinical interview. Subsequent studies have validated the EPDS for use in the perinatal period (during pregnancy and postpartum). The Edinburgh Postnatal Depression Scale (EPDS) is a tool comprising 10 items created specifically to screen for peripartum depressive symptomatology. Its use for the detection depressive symptoms in both antepartum and postpartum samples has been extensively adopted [52,53,54]. Items are scored from 0 to 3 while total scores range from 0 to 30 with higher scores reflecting a more severe form of depression. We administered the Italian language version of EPDS which constitutes a valid tool for detecting maternal antenatal and postnatal depressive symptomatology with acceptable internal consistency and test–retest reliability [49,55]. In numerous validation studies, cut-off points have varied. It is important to recognize that the optimal EPDS cut-off scores change considerably during different phases of the peripartum period [56,57]. Many international studies have reported an optimal cut off score of the ≥12/13 in late pregnancy [51,57,58]. However, as suggested by Dennis [59], a cut-off score ≥ 9 is normally used in order to identify patients with depressive symptomatology in the early post-partum period [48,59,60]. Therefore, in this paper, we choose to use EPDS cut-off scores of ≥12 and ≥9, respectively, in the assessment for antepartum depressive symptomatology and for depressive symptoms in the immediate postpartum period; our aim is to make the findings comparable with both local and international studies. The EPDS t0 and t1 showed internal consistency (Cronbach’s alpha values were 0.78 and 0.79, respectively).

### 2.2. Experiences in Close Relationship Scale

The Experiences in Close Relationship Scale (ECR) is one of the most used self-report instruments for the assessment of adult attachment and has been widely adopted in previous studies assessing attachment as a risk factor for antenatal and postpartum depression [23,61,62]. The ECR comprises 36 items, and they are scored on a seven-point Likert scale, with 1 corresponding to “disagree strongly” and 7 corresponding to “agree strongly”. It is composed of two subscales, namely attachment anxiety (concerning rejection or abandonment), and attachment avoidance (of intimacy and interdependency in close relationships) [60]. We employed the Italian version of the ECR [63], which was used in previous Italian studies on the relationship between attachment patterns and perinatal depression [64,65]. In the current study, the internal consistency was good (Cronbach’s alpha 0.78). 

### 2.3. Coping Orientation for Problem Experiences

Coping strategies were assessed with the brief Coping Orientation for Problem Experiences (COPE) which consists of 14 scales: active coping, planning, behavioral disengagement, self-distraction, seeking emotional support, seeking instrumental support, venting of emotions, positive reframing, humor, acceptance, denial, religion, substance use, and self-blame [42]. The model developed by Carver et al. [37] consists of three types of coping strategies: emotion-focused coping, problem-focused coping, and avoidant coping. Problem-focused coping strategies that are aimed at changing the stressful situation include active coping, seeking instrumental support, planning, and positive reframing. Emotion-focused coping strategies that are aiming to regulate emotions associated with the stressful situation include venting of emotions, seeking emotional support, humor, acceptance, self-blame, and religion. The third type of coping, avoidance-focused coping, which indicates physical or cognitive efforts to disengage from the stressors, includes self-distraction, denial, substance use, and behavioral disengagement [66,67]. We used the Italian language version developed by Conti [68]. This scale was previously used in a sample of Italian mothers to study the link between coping and postpartum depressive symptoms [69]. The internal consistency of the total Brief COPE was high (Cronbach’s was alpha 0.85).

### 2.4. Data Analysis

Statistical analyses were performed using the IBM Statistical Package for Social Sciences (SPSS), version 26.0, and AMOS, version 22.0. The data were scanned for missing values and possible multivariate outliers via the Mahalanobis distance, with a criterion of *p* < 0.001 to remove the participants. The skewness and kurtosis were used to examine the distribution of the data. The linear association between attachment styles and stress coping strategies was examined by Pearson’s correlation. We used logistic regression analysis to evaluate the relationship between independent variables including medications, medical condition during pregnancy, complications during pregnancy, and family history (FH) of psychiatric disorders on binary dependent variables (t0: EPDS ≥ 12 or t1: EPDS ≥ 9), respectively, in two different periods: pregnancy and postpartum.

Inflation variance (VIF) and tolerance factors were used to assess multiple alignment between variables. We conducted a second-order exploratory factor analysis (EFA) on the subscales of the Brief COPE in two different periods (pregnancy and postpartum) in order to reduce the number of factors and create a more specific model of coping strategies. A confirmatory factor analysis (CFA) was performed to validate the results of the EFA. Our main hypotheses were tested using structural equation modelling (SEM) to assess the mediating role of the different higher-order coping strategies in the relationship between attachment and perinatal depression at t0 and t1. Following the recommendations of Schreiber et al. [70], we used the criteria of the Comparative Fit Index, the Root Mean Square Error of Approximation (RMSEA), the Standardized Root Mean Square Residual (SRMR), and the minimum discrepancy per degree of freedom (CMIN/DF) to analyze the fit of the models. As recommended in SEM literature [71], alternative models were also tested to eliminate alternative explanations. To confirm the presence of mediation, bootstrapping, with 5000 bootstrap samples, was used to calculate the total, direct, and indirect effects of the predictors (attachment styles) on the outcome variables (perinatal depression). The total, direct, and indirect effects are reported as estimates with 95% confidence intervals. The level of statistical significance was set at *p* < 0.05.

## 3. Results

According to the EPDS score ≥ 12, 14.1% of the women (234/1664) suffered from depressive symptomatology in pregnancy, and according to the EPDS score ≥ 9, 18.5% (309/1664) in the postpartum period. Additionally, 19.4.6% (60/309) of women who had antepartum depressive symptoms were found to have postpartum depressive symptomatology. Missing values and cases found to be outliers were excluded from the analyses, leaving the antepartum total sample in 234 and the postpartum total sample in 287/1351 participants. The descriptive statistics of the study samples are illustrated in Table 1.

Differences between women with and without major depressive symptoms according to the EPDS scores in all analyzed variables in the antepartum and postpartum phase are reported in Table 2. Therefore, women with depression scored significantly higher in measurements of anxiety and avoidant attachment. Coping strategies of adaptive value, most frequently observed in women without depression, were those focused on acceptance of their situation, during both the prenatal and postnatal stages. The coping strategies that were most frequently associated with depression were denial, venting of emotion, self-blame, seeking emotional support, and behavioral disengagement.

The results of Pearson’s correlations between attachment styles and coping strategies are reported in Table 3. The highest correlation was between avoidant attachment and anxious attachment. In the prenatal phase, anxious attachment was also positively correlated with seeking emotional support and dysfunctional coping strategies such as denial, self-blame, venting of emotion, and behavioral disengagement; avoidant attachment was positively correlated with seeking instrumental support, denial, behavioral disengagement, and substance use. An examination of coping strategies during the postnatal stage also provides substantial insights. Anxious attachment was positively correlated with seeking emotional support, seeking instrumental support, venting of emotion, denial, behavioral disengagement, self-blame, religion, and substance use; avoidant attachment was correlated with behavioral disengagement, denial, and substance use. 

Logistic regression analyses were used to examine the role of risk factors on depressive symptomatology. The first logistic regression showed that family history of psychiatric disorders was related to EPDS ≥ 12 during pregnancy (β = 0.8216, *p* < 0.001) and to EPDS ≥ 9 (β = 0.7043, *p* < 0.001) in the postpartum period. Furthermore, additional t-test analyses showed that significant differences in coping strategies between women with family history and women without family history of psychiatric disorders. Women with a FH of mental illness exhibited higher utilization of maladaptive avoidant coping strategies (i.e., behavioral disengagement, denial, self-distraction, and substance use) compared with women without FH of mental disorders during either pregnancy or post-partum period (all, *p* < 0.05).

### 3.1. Exploratory Factor Analysis

An EFA on the subscales of the Brief COPE was performed in two different periods (pregnancy and postpartum) using a principal component analysis and varimax rotation. Eigenvalues of 1.0 were used to determine the number of components extracted. The minimum factor loading was set to 0.50 [72]. The factor solution derived from the EFA in participants with antenatal depressive symptomatology yielded three distinct second-order factors, which accounted for 59.048 percent of variation in the data. No items showed cross-loadings of 0.50 or above on more than one factor. Barlett’s test of Sphericity, which indicates whether the correlations in the data are sufficiently strong to use a dimension-reduction technique such as principal components analysis, was significant (χ^2^ (55) = 657.564; *p* < 0.001) [73]. The Kaiser–Meyer–Olkin (KMO) measure of sampling adequacy, which is a measure of the suitability of the data for factor analysis, was 0.719, which was well above the acceptable threshold of 0.50 [74,75]. According to the factor loading criteria, the subsequent factors comprised the following sub-categories: Factor 1, adaptive coping, includes the scales of active coping, planning, positive reframing, acceptance, and humor. Factor 2, emotional coping, includes the scales of seeking instrumental support, seeking emotional support, venting of emotions, and self-blame. Factor 3, maladaptive coping, includes the scales of denial, behavioral disengagement, and substance use. No items cross-loaded on multiple factors with loadings greater than 0.50. Self-distraction and religion did not load above 0.3 in any of the three predicted factors. These scales were therefore left out of the analyses. Similar results were found when confirmatory factor analysis was conducted on participants with postpartum depressive symptomatology. The factor structure derived from the EFA yielded three distinct second-order factors, which accounted for 57.541 per cent of variation of the data. The Barlett’s test of Sphericity was significant (χ^2^ (66) = 450.237, *p* < 0.001). The KMO was 0.65. The resulting factors comprised the following sub-scales: Factor 1, adaptive coping, includes the scales of positive framing, acceptance, planning, and humor. Factor 2, emotional coping, includes the scales of seeking instrumental support, seeking emotional support, venting of emotions, and self-blame. Factor 3, maladaptive coping, includes the scales of denial and behavioral disengagement. No items cross-loaded on multiple factors with loadings greater than 0.50. Self-distraction and religion and active coping did not load above 0.3 in any of the three predicted factors. Therefore, these scales were excluded from the analyses.

### 3.2. Confirmatory Factor Analysis

In order to determine the reliability of the factor structure derived from the EFA, a CFA was used. An initial test of the three-factor model, derived from the EFA in participants with antepartum depression, indicated a less than acceptable fit with the following parameters: CMIN/DF = 3.027; CFI = 0.831, RMSEA = 0.098, SRMR = 0.082. The examination of the standardized residuals and modification indices suggested correlation of errors between planning and active coping; reframing and humor; venting and self-blame; instrumental support and self-blame; and denial and substance. This new model resulted in CMIN/DF = 2.388, CFI = 0.908, RMSEA = 0.07, and SRMR = 0.064, meeting all the fitting criteria. The definitive three-factor model reached a much better fit than the three-factor model that distinguishes problem-focused coping, emotion-focused coping, and avoidant coping [37]: CMIN/DF = 4.559, CFI = 0.63, RMSEA = 0.125, and SRMR = 0.114. Similar results were obtained testing the three-factor model derived from the EFA in participants with postpartum depressive symptoms. The initial test indicated a poor fit: CMIN/DF = 3.319, CFI = 0.804, RMSA = 0.086, SRMR = 0.078. Modification indices suggested adding covariances between seeking instrumental support and seeking emotion, seeking instrumental support and self-blame, humor and acceptance, and humor and positive reframing. The model reached an excellent fit index: CMIN/DF = 2.040, CFI = 0.95, RMSEA = 0.059, SRMR = 0.053. This definitive model showed a much better fit than the three-factor model that distinguishes problem-focused coping, emotion-focused coping, and dysfunctional coping: CMIN/DF = 3.632, CFI = 0.71, RMSEA = 0.106, and SRMR = 0.109.

### 3.3. SEM Analysis

Before undertaking the analysis, the assumptions of SEM including normal distribution and multiple alignment were examined. We assessed the distribution of all variables and transformed those variables that showed strong non-normality. Log transformation was used for EPDS postpartum and substance use scores. All the other variables were normally distributed [76]. There was no alignment between the variables (VIF amplitude was less than 10, and tolerance was higher than 0.1). To analyze the mediation effects of coping strategies on the relationship between attachment styles and peripartum depressive symptomatology, the three second-order factors established using the CFA were used in the structural equation modelling (SEM) analysis. Because of the complexity of the models, only the higher-order coping strategies were estimated as latent factors. The factor scores were utilized as manifest variables for all other constructs in the model. The logistic regression analyses used to examine the role of risk factors on depressive symptomatology showed a significant association between FH of psychiatric disorders and perinatal depressive symptomatology; thus, this control variable was included in the SEM models.

#### 3.3.1. SEM Antepartum Depressive Symptomatology

The first model, which investigated the importance of coping strategies in mediating the relationship between attachment styles and antepartum depressive symptomatology, consisted of two independent variables (avoidance attachment and anxiety attachment), three intermediate variables (adaptive, emotional, and maladaptive coping), and the dependent variable (antepartum mood symptoms) based on EPDS score ≥ 12. The first model did not have any acceptable fit indices at this point (CMIN/DF = 7.81, CFI = 0.800, RMSEA = 0.097, SRMR = 0.103). Non-significant paths (attachment avoidant path to the maladaptive coping and the path of anxious attachment to ante-partum depressive symptomatology and to active coping) were eliminated stepwise to create a more parsimonious model. Then, based on modification indices, we allowed covariation of the error terms of instrumental support and blame, instrumental support and emotional support, substance use and denial, humor and reframing, and planning and active coping. This new model reached acceptable fit indices (CMIN/DF = 2.19, CFI = 0.894, RMSEA = 0.071, SRMR = 0.065) [72,77]. All regressions were statistically significant (*p* < 0.01). A diagrammatic representation of the structural models is presented in Figure 1. Results showed that anxious attachment was positively associated with both maladaptive coping and emotive coping strategies. Avoidant attachment was associated with ante-partum depressive symptomatology and less emotive and active coping strategies. Emotional and maladaptive coping strategies were positively associated with antepartum depressive symptoms, while active coping strategies were negatively associated with antepartum depressive symptoms. To further confirm the specificity of coping strategies as mediators of these relations, based on existing literature findings [78], an alternative model was also tested whereby antepartum depressive symptomatology was included as the mediator variable and coping strategies as the outcome variables. This model was nonsignificant (CMIN/DF = 3.224, CFI = 78.1, RMSEA = 0.098, SRMR = 0.0874). In summary, structural equation modelling identified a significant mediational effect of emotional and maladaptive coping strategies on the relation between anxiety attachment and antepartum depressive symptomatology. Results showed a significant mediational effect of emotional coping strategies on the relationship between avoidant attachment and ante-partum depressive symptoms. Results from bootstrapping analysis confirmed a mediating effect of emotional and maladaptive coping strategies on the relationships between anxious attachment and ante-partum depressive symptomatology (two-tailed significance bias-corrected, *p* = 0.005; 95% confidence interval 0.01 to 0.04). Moreover, results confirmed a mediating effect of emotional coping strategies on the relationships between avoidant attachment and antepartum depressive symptomatology (95% confidence interval −0.03 to −0.04; two-tailed significance bias-corrected, *p* = 0.002).

#### 3.3.2. SEM Postpartum Depressive Symptomatology

The second model, which investigated the mediating role of coping strategies between attachment styles and postpartum depressive symptomatology, consisted of two independent variables (avoidance attachment and anxiety attachment), three intermediate coping factors (adaptive, emotional, and passive coping), and the dependent variable (postpartum mood symptoms) based on EPDS score ≥ 9. An initial test of the model resulted in poor model fit (CMIN/DF = 3.598, CFI = 0.805, RMSEA = 0.095, SRMR = 0.073). In order to obtain a satisfactory model fit, all nonsignificant pathways were sequentially removed from the initial model. This was done to evaluate a simpler, more parsimonious model that contained fewer pathways. The investigation of modification indices suggested adding covariances between humor and acceptance, humor and planning, blame and venting, and seeking instrumental support and seeking emotional support. Results of the final model revealed a satisfactory fit (CMIN/DF = 2.676, CFI = 0.874, RMSEA = 0.077, SRMR = 0.650). All regressions were statistically significant (*p* < 0.01). Figure 2 shows the path coefficients. Anxious attachment style was positively associated with emotive coping and maladaptive coping and negatively associated with active coping. Avoidant attachment style was associated with maladaptive coping. The relationships between emotional coping, active coping, and postpartum depressive symptomatology were comparable to those found in the previous antepartum model. Results showed that emotional coping strategies were positively associated, and active coping strategies were negatively associated with post-partum depressive symptoms, respectively. Maladaptive coping was not associated with post-partum depressive symptoms, indicating that it did not serve any significant mediating role in the relationship between attachment styles and post-partum depression. The alternative model with post-partum depression and coping strategies swapped (with post-partum as mediator and coping strategies as the dependent variables) was nonsignificant (CMIN = 3.821, CFI = 0.784, RMSA = 0.099, SRMR = 0.945). In summary, structural equation modelling identified a significant mediational effect of emotional coping and less active coping strategies on the relationship between anxiety attachment and post-partum depressive symptomatology. Results from bootstrapping analysis confirmed a mediating effect of emotional and active coping strategies on the relationships between anxious attachment and post-partum depressive symptoms (95% confidence interval 0.0 to 0.01; two-tailed significance bias-corrected, *p* = 0.005).

## 4. Discussion

This study was the first to test a model in which the relationship between attachment styles and depressive symptomatology is mediated by coping strategies in both antepartum and postpartum periods. The main result of the study is that attachment characteristics had a greater predictive power on depressive symptoms when its direct and indirect statistical effects are taken into consideration. As expected, based on evidence [4,26,27,79], preliminary analyses showed strong correlations between independent variables (avoidance attachment and anxiety attachment), intermediate variables (coping strategies), and the dependent variables (perinatal depression outcomes), laying the groundwork for mediation analysis [80]. First, results supported a three-factor structure of the coping strategies using the Brief COPE questionnaire, with the identification of three groups of coping strategies (i.e., active coping, maladaptive coping, and emotional coping) in both antepartum and postpartum samples [42]. In accordance with the studies by Carver [42] and Farley et al. [81], concerning the Brief COPE, the higher-order factor that we denominated emotional coping encompassed seeking emotional and instrumental support. Second, we examined the role of these three high-order factors of coping in mediating the relationship between attachment dimensions and antenatal or postnatal depressive symptoms. The results of SEM analyses revealed that the hypothesized models of this study had a good fit in the study samples. Despite the fact that previous studies have consistently found linear associations between dimensions of insecure attachment styles and peripartum depressive symptomatology [5], here we show that coping strategies play a significant role in mediating the effect of anxious and avoidant attachment styles on depressive symptomatology. Moreover, we show that different dimensions of attachment styles and dysfunctional coping strategies were associated with either antepartum or postpartum depressive symptoms. In line with previous studies, the results indicated that, besides the direct effect of avoidant attachment on antepartum symptoms, the effect of insecure attachment styles (anxious and avoidant attachment) on antepartum depressive symptomatology was partially mediated by emotional coping styles. Anxious attachment also has an indirect significant effect on postpartum depression through emotional coping; notwithstanding, it seems that avoidant attachment is not significantly associated with postpartum depressive symptoms [82,83,84,85].

## 5. Antepartum Depressive Symptomatology

The results of mediational analyses revealed that anxious attachment style was significantly associated with antepartum depressive symptomatology through emotional and maladaptive coping strategies. These findings are consistent with prior studies that suggest the connection between anxious attachment and psychological distress, encompassing depressive symptoms, is not a straightforward relationship; instead, coping has an important function in mediating this relationship [28,30]. Regarding relations between anxious attachment and coping strategies, the present study shows that anxiously attached individuals use more ineffective maladaptive coping strategies (i.e., behavioral disengagement, denial, substance use) and emotional coping strategies (i.e., seeking instrumental support, seeking emotional support, venting of emotions, self-blame) compared to the effective ones. The findings are consistent with previous studies showing that those individuals with anxious attachment use emotion-focused and maladaptive coping strategies more frequently when dealing with stresses [82,83,84]. As emotion-focused coping strategies are most often used when the individuals perceive the stressor as something that cannot be altered and must be endured, we assume that women with anxious attachment utilized these strategies to lessen the emotional distress associated with the pregnancy and post-partum [6]. Additionally, as people with attachment anxiety reported high levels of alienation and tend to be withdrawn, these individuals might also rely to a greater extent on a variety of coping strategies, including dysfunctional coping such as maladaptive coping strategies [85].

Avoidant attachment style showed both a direct effect on antenatal depressive symptoms and an indirect effect through emotional coping. Our findings are consistent with those of a previous study showing that attachment avoidance has a direct effect on measures of psychological distress as well as an indirect effect via ineffective problem coping styles [30]. Early studies reported that attachment avoidance does not significantly predict distress when attachment anxiety was controlled for [28,29]. However, our results, consistent with previous research [30], showed that attachment avoidance predicted antenatal depressive symptomatology even when controlling for the other attachment dimension. In considering the relationship between attachment and coping, the findings of the present study indicate a negative correlation between avoidant attachment style and emotional coping. As several emotional coping strategies (e.g., self-blame and venting of emotions) are intended to relieve the psychological distress without necessarily engaging with and relying on others for support, the observed negative correlation was unexpected [37]. However, those with attachment avoidance are inclined to a lesser recognition of distress and skeptical of trust in relationships. Thus, as previous research as demonstrated, the negative correlation between emotional coping and attachment avoidance could be attributable to the negative correlation between attachment avoidance and perceived stress as well as to the difficulty in expressing their feelings to others [86,87].

## 6. Postpartum Depressive Symptomatology

Our findings indicate that emotional coping and less use of active coping strategies mediate the relations between anxious attachment and postpartum depressive symptomatology, while the effects of avoidant attachment were not significant. The results are consistent with previous studies [4,5] showing that anxious style was found to be associated with postpartum depressive symptomatology more frequently than avoidant style of attachment. Similarly, in the relevant literature, significant and positive correlations exist between anxious attachment and perceived stress. However, it seems that avoidant attachment is not significantly related to stress symptoms [88,89,90,91]. This is in line with the observations, as discussed above, that persons with attachment avoidance tend not to acknowledge their distress [83,92]. The fact that avoidant attachment style had a significant and direct effect on antepartum depressive symptoms but did not have any significant effect on postpartum depressive symptomatology may be indicative of the heterogeneous nature of the avoidant attachment style which encompasses both a positive and negative sense of self. It is also possible that those with avoidant attachment experiencing a greater level of perceived stress during postpartum period reported greater utilization of avoidant coping strategies to distance themselves from actively processing and/or resolving the distress. Indeed, it is noteworthy that avoidant attachment style was strongly correlated with maladaptive coping, including self-distraction, behavioral disengagement, and denial. Given that prior research has demonstrated a significant association between maladaptive coping and perinatal depressive symptomatology [3,11], it was expected that these coping strategies would mediate the relationship between avoidant attachment and post-partum depressive symptomatology. One possible reason for this inconsistency is that structural equation modelling used in the present study is potentially more powerful and less susceptible to bias than the regression methods used in other studies. These results may also be due to a stronger conceptualization and measurement of coping strategies.

The findings regarding anxious attachment are in line with the results reported in the literature, showing a strong association between anxious attachment styles with postnatal depressive symptoms [4,5]. We extended these original observations by providing evidence that anxious attachment has an indirect significant effect on postpartum depressive symptoms through ineffective coping strategies (e.g., emotional coping). These results confirm analyses reported in other earlier studies revealing patterns of the relationship between anxious attachment styles and ways of coping with difficult situations [28,30].

There are several shortcomings in this study that are noteworthy. First, data were collected through self-assessment questionnaires, which might reduce the validity of the results. Even so, the questionnaires we used in the current study were reliable and commonly used research instruments. Moreover, although a validated measure was used to assess attachment styles, the study could have been strengthened by using multiple attachment measures to operationalize insecure attachment styles. Second, although there has not yet been an Italian validation study for the use of EPDS in pregnancy, we used EPDS to assess antenatal depressive symptomatology. However, according to previous Italian studies, which investigated the prevalence and determinants of antenatal depression among Italian mothers, we used a cut-off > 12 [54,93,94]. Another limitation of the study was the absence of a clinical interview; thus, we could not directly compare EPDS scores with the clinical diagnosis of major depression. Third, the limitations imposed by the cross-sectional design of this study curtailed its capacity to comprehensively analyze the association between attachment styles and perinatal depression outcomes as mediated by coping strategies. Longitudinal studies could provide a more rigorous evaluation of these associations. Fourth, even though the results are derived from SEM, which is a more intricate model than those typically utilized in existing literature, they are still based on correlational data. Hence, it prevents us from forming any definitive conclusions about the cause-and-effect relationships between the variables. Fifth, our data collection was made in the first months of the first wave of the COVID-19 pandemic. Several previous studies specifically exploring the impact of COVID-19 on maternity, during the first severe wave of the COVID-19 pandemic (2020), reported a significant higher prevalence and levels of depressive symptomatology in Italian mothers during the pandemic than previously reported in literature [69,95]. However, results have been inconsistent, and other findings did not show significant differences in depression and anxiety symptoms between women who gave birth before and during the pandemic period after the implementation of COVID-19 restrictions and threats, suggesting that reproductive experience alters the female brain in adaptive ways [96]. We cannot rule out the possibility that our findings have been influenced by high levels of perceived stress among pregnant women during COVID-19. Unfortunately, we did not measure pandemic-related stress symptoms. However, it should be noted that women were likely to use avoidant coping strategies like substance abuse and self-distraction for dealing with the stressful COVID-19-related situation [69]. Instead, in our sample, women with insecure attachment styles chose emotional coping strategies more frequently. More research is needed to clarify the impact of COVID-19 on coping strategies and depressive symptomatology during and after pregnancy.

Finally, our results, in line with previous research, revealed a significant association between family history of any psychiatric illness and maternal peripartum depressive symptomatology [97]. Additionally, we found a significative difference in maladaptive coping strategies between women with family history of mental illness compared with women without. These findings might suggest that risk for peripartum depressive symptomatology may be importantly influenced by the interaction between family history of mental illness and maladaptive coping strategies. However, our results showed that coping strategies play a significant role in mediating the effect of attachment styles on depressive symptomatology even when controlling for the family history of psychiatric disorders. Further research is needed to explore the potential implications of the relationship between FH, coping, and perinatal depressive symptomatology for developing targeted interventions and support strategies for women with family history of mental disorders.

## 7. Conclusions

Despite these limitations, our findings suggest that insecure attachment styles associated with a greater use of ineffective coping strategies can meaningfully predict perinatal depressive symptomatology. Particularly, our results suggest that different patterns of attachment and related coping strategies were associated with depressive symptoms during different phases of the peripartum period. We found a significant indirect effect between attachment anxiety and maternal depressive symptoms via emotional and maladaptive coping strategies during pregnancy and via emotional coping strategies during the early postpartum period. Avoidant attachment style showed both a direct effect and an indirect effect through emotional coping on antenatal depressive symptoms, whereas no significant effects involving ineffective coping strategies on the pathway from attachment avoidance to depressive symptomatology were found during the postpartum period. The implications of these findings suggest that a better understanding of dynamics of women’s attachment and chosen coping strategies symptoms in different stages of the peripartum period may be useful in identifying and addressing risk factors associated with maternal depressive symptomatology. Further research would allow an exploration of the relationships between attachment and stress reactions as mediated by coping strategies. This would help a preventive coping-based intervention strategy to enhance the capacity of women with insecure attachment styles (especially anxious attachment) to use more adaptive coping during and after pregnancy.

## Figures and Tables

**Figure 1 brainsci-13-01002-f001:**
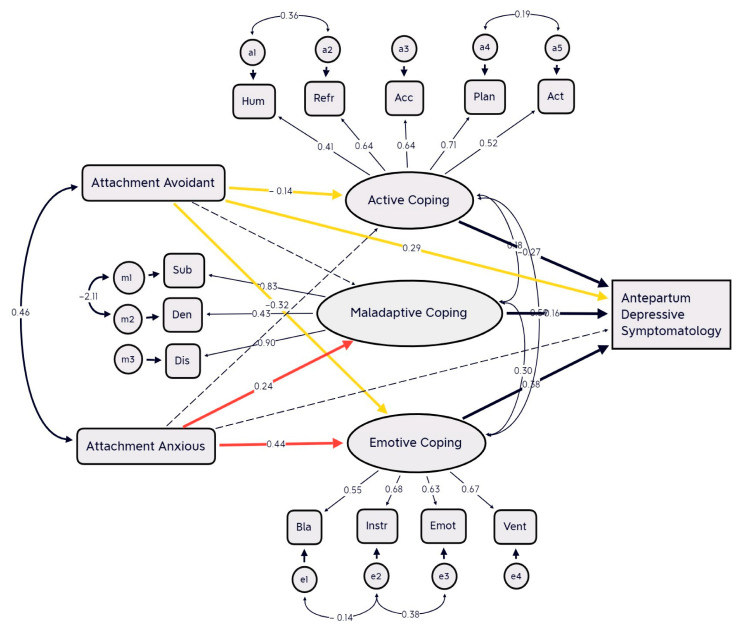
Path analysis model of the mediating role of coping in the relationship between attachment styles and antepartum depressive symptomatology. Values represent standardized path coefficients, factor loadings, and covariance estimates for the tested model. Solid lines represent significant effects (*p* < 0.05), and dashed lines represent non-significant effects (*p* < 0.05). The colored and bold lines represent significant pathways. Hum = humor; Refr = reframing; Acc = acceptance; Plan = planning; Act = active coping; Sub = substance use; Den = denial; Dis = behavioral disengagement; Bla = self-blame; Instr = seeking instrumental support; Emot = seeking emotional support; Vent = venting of emotions.

**Figure 2 brainsci-13-01002-f002:**
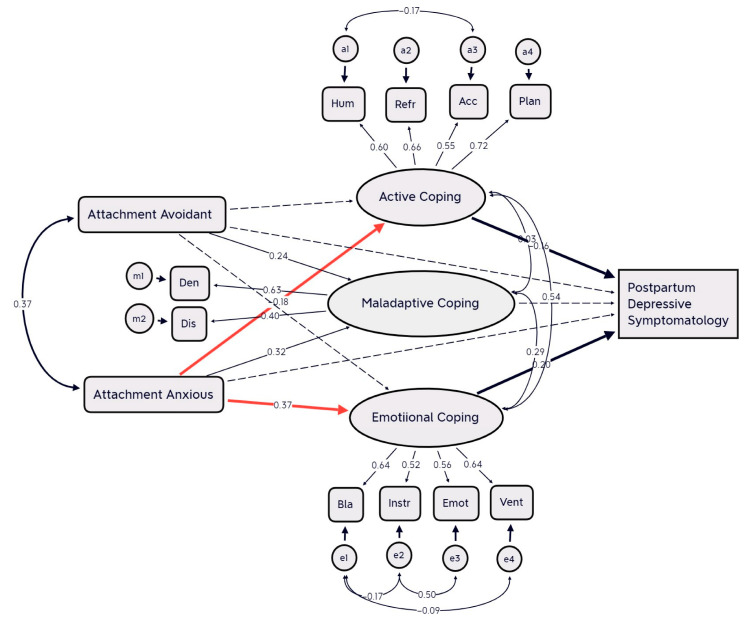
Path analysis model of the mediating role of coping in the relationship between attachment styles and postpartum depressive symptomatology. Values represent standardized path coefficients, factor loadings, and covariance estimates for the tested model. Solid lines represent significant effects (*p* < 0.05), and dashed lines represent non-significant effects (*p* < 0.05). The colored and bold lines represent significant pathways. Hum = humor; Refr = reframing; Acc = acceptance; Plan = planning; Act = active coping; Den = denial; Dis = behavioral disengagement; Bla = self-blame; Instr = seeking instrumental support; Emot = seeking emotional support; Vent = venting of emotions.

**Table 1 brainsci-13-01002-t001:** Characteristics of women with either antepartum or early postpartum depressive symptomatology.

	Antepartum (*N* = 234)	Postpartum (*N* = 287)
Age (mean, SD)	32.2 (5.9)	32.9 (5.2)
Educational Level (%)		
Primary school	1 (0.42)	1 (0.34)
Secondary school	37 (15.8)	34 (11.8)
Post-Secondary school	116 (49.5)	155 (54.0)
Higher education	79 (33.7)	96 (33.4)
Parity (primiparous) (%)	111 (47.4)	151 (52.6)
Marital status (%)		
Married or paired	230 (98.2)	286 (99.6)
Some medical condition		
during pregnancy (%)	64 (27.3)	85 (29.6)
Complication during pregnancy (%)	154 (65.8)	278 (96.8)
Family history of psychiatric		
problems (%)	147 (62.8)	141 (49.1)
Medications		
Corticosteroid	2 (0.85)	5 (1.74)
Insulin	21 (8.97)	25 (8.71)
Levothyroxine	26 (11.1)	38 (13.2)
Antihypertensives	1 (0.42)	10 (3.48)
Anticoagulants	8 (3.41)	10 (3.48)
Anxiolytics	13 (5.55)	13 (4.52)
Others	14 (5.98)	14 (4.87)

**Table 2 brainsci-13-01002-t002:** Coping strategies in women with or without postpartum depressive symptomatology. Mean (SD).

	Antepartum	Antepartum	Postpartum	Postpartum
	w/Depressive	w/o Depressive	w/Depressive	w/o Depressive
	Symptoms	Symptoms	Symptoms	Symptoms
	(N = 234)	(N = 1430)	(N = 287)	(N = 1351)
Age	32.2 (5.95)	32.4 (5.43)	32.9 (5.17)	32.3 (5.54)
EPDS	14.4 (2.57)	5.17 (3.23) ***	11.7 (2.93)	3.98 (2.15) ***
Anxious Attachment	57.2 (21.6)	39.3 (16.7) ***	52.0 (21.0)	39.3 (16.8) ***
Avoidant Attachment	38.7 (18.0)	28.5 (12.1) ***	33.7 (15.1)	28.8 (12.7) ***
Coping Strategies				
Active Coping	6.38 (6.71)	6.71 (1.38) *	6.53 (1.34)	6.71 (1.38)
Seeking informational support	5.73 (1.60)	5.31 (1.59) **	5.52 (1.53)	5.34 (1.59)
Planning	6.12 (1.439)	6.45 (1.46) ***	6.29 (1.31)	6.44 (1.47)
Positive Reframing	5.73 (1.51)	6.20 (1.45) **	6.01 (1.31)	6.17 (1.48)
Venting of emotions	5.46 (1.509	4.84 (1.73) **	5.25 (1.43)	4.86 (1.75) ***
Seeking emotional support	5.58 (1.64)	4.84 (1.01) **	5.28 (1.61)	4.87 (1.63) ***
Humor	4.00 (1.53)	4.29 (1.33) *	4.61 (1.38)	4.29 (1.35)
Acceptance	6.06 (1.42)	6.27 (1.34) *	6.04 (1.27)	6.28 (1.35) **
Self-blame	5.76 (1.48)	4.95 (1.43) **	5.52 (1.42)	4.96 (1.44) ***
Religion	5.12 (2.05)	4.87 (2.00)	5.21 (1.91)	4.85 (2.01) **
Self-distraction	5.67 (1.54)	5.20 (1.61) **	5.51 (1.47)	5.22 (1.62) **
Denial	4.05 (1.69)	3.11 (1.33) **	3.68 (1.46)	3.14 (1.39) **
Substance use	2.21 (0.87)	2.05 (0.43) *	2.03 (0.21)	2.07 (0.52)
Behavioral disengagement	3.65 (1.57)	2.81 (1.16) **	3.27 (1.35)	2.83 (1.20) ***

EPDS = Edinburgh Postnatal Depression Scale. * *p* < 0.05. ** *p* < 0.01. *** *p* < 0.001.

**Table 3 brainsci-13-01002-t003:** Correlation coefficients between attachment styles and stress coping strategies.

	Antepartum		Postpartum	
	(*N* = 234)		(*N* = 287)	
	Anxious	Avoidant	Anxious	Avoidant
Anxious Attachment		0.463 **		0.373 **
Avoidant Attachment	0.463 **		3.73 **	
Active Coping	−0.002	−0.186 **	0.044	−0.073
Positive Reframing	−0.126	−0.120	−0.208 **	−0.199 **
Planning	−0.015	−0.032	−0.073	−0.149 *
Humor	−0.101	−0.046	−0.095	0.018
Acceptance	−0.129 *	−0.064	−0.168 *	−0.128 *
Seeking emotional support	0.228 *	−0.091	0.321 **	0.051
Seeking instrumental support	0.043	−0.201 **	0.167 **	−0.111
Behavioral disengagement	0.225 **	0.223 **	0.117 *	0.202 **
Self-distraction	0.023	−0.015	0.077	0.102
Self-blame	0.270 **	−0.078	0.209 **	0.090
Venting of emotions	0.188 **	−0.047	0.244 **	0.056
Denial	0.231 **	0.143 *	0.238 **	0.197 **
Religion	0.083	−0.103	0.125 *	0.003
Substance use	−2.22 **	0.135 *	0.158 *	0.169 **

* *p* < 0.05; ** *p* < 0.01 (two tailed).

## Data Availability

The data presented in this study are openly available in FigShare at https://doi.org/10.6084/m9.figshare.23100746, accessed on 23 May 2023.

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
