# Peer review of "Coping as a Mediator between Attachment and Depressive Symptomatology Either in Pregnancy or in the Early Postpartum Period: A Structural Equation Modelling Approach"

_brainsci, 2023, doi:10.3390/brainsci13071002_

Round 1

Reviewer 1 Report

Altamura et.al. showed the role of coping as a mediator between attachment and depression during pregnancy or in the early postpartum period. Overall this manuscript is well written and quite readable. The analyses are all performed properly. The authors also list the limitation of this study in the Discussion section. Here are my comments.

1. Did the authors consider whether these women used any medications? Certain drugs are known for causing mental symptoms.

2. The title of Table 1 is misleading. I think the authors are showing two groups of women. The title makes me think that there is only one group of women with both antepartum and postpartum depression.

3. It seems that a large number of women with depression have a family history of psychiatric problems (62.8 percent and 49.1 percent, respectively). This is a strong factor. I would suggest that the authors look at how the strategies for coping with stress work for these women in comparison to other women.

4. It is also worthy to add some discussion about the relation between family history of psychiatric problems and depression.

5. What the numbers in Fig .1 and 2? It needs some description in the figure legend.

6. Add some colors in the figures will make them easier to read.

Author Response

Reviewer # 1

Altamura et.al. showed the role of coping as a mediator between attachment and depression during pregnancy or in the early postpartum period. Overall this manuscript is well written and quite readable. The analyses are all performed properly. The authors also list the limitation of this study in the Discussion section. Here are my comments.

  1. Did the authors consider whether these women used any medications? Certain drugs are known for causing mental symptoms.

We have now included the variable “medications” in  tab. 1 and conducted additional analyses.

Tab. 1 Characteristics of women with either antepartum or early postpartum depressive symptomatology

            Antepartum                                           Postpartum

              (N=234)                                               (N=287)

Age (mean, SD)                                                                   32.2 (5.9)                                               32.9 (5.2)                              

Educational Level (%)                                       

     Primary school                                                               1     (0.42)                                              1     (0.34)

     Secondary school                                                           37   (15.8)                                              34   (11.8)

     Post-Secondary school                                                 116 (49.5)                                              155 (54.0)

     Higher education                                                           79   (33.7)                                              96   (33.4)

Parity (primiparous) (%)                                                    111 (47.4)                                              151 (52.6)

Marital status (%)

     Married or paired                                                           230 (98.2)                                              286 (99.6)                           

Medical condition

during pregnancy (%)                                                         64   (27.3)                                              85   (29.6)

Complication during pregnancy (%)                                154 (65.8)                                              278 (96.8)

Family history of psychiatric

disorders (FH) (%)                                                              147 (62.8)                                              141 (49.1)

Medications

     Corticosteroid                                                                2     (0.85)                                              5    (1.74)

     Insulin                                                                              21   (8.97)                                              25  (8.71)

     Levothyroxine                                                26   (11.1)                                              38  (13.2)

     Antihypertensives                                                          1     (0.42)                                              10  (3.48)

     Anticoagulants                                                               8     (3.41)                                              10  (3.48)

     Anxiolytics                                                                     13   (5.55)                                              13  (4.52)

     Others                                                                               14   (5.98)                                             14  (4.87)             

We have now added the following phrases in the analysis section and results section.

“We used logistic regression analysis to evaluate the relationship between independent variables including medications, medical condition during pregnancy, complications during pregnancy and family history (FH) of psychiatric disorders on binary dependent variables (t0: EPDS ≥ 12 or t1: EPDS ≥ 9) respectively, in two different periods: pregnancy and postpartum.”

“Logistic regression analyses were used to examine the role of risk factors on depressive symptomatology. The first logistic regression showed that family history of psychiatric disorders was related to EPDS ≥ 12 during pregnancy (β=0.8216, p<0.001) and to EPDS ≥ 9 (β= 0.7043, p<0.001) in the postpartum period. Furthermore, additional t-test analyses showed that significant differences in coping strategies between women with family history and women without family history of psychiatric disorders. Women with FH of mental illness exhibited higher utilization of maladaptive avoidant coping strategies (i.e., behavioral disengagement, denial, self-distraction, and substance use) compared with women without FH during either pregnancy or post-partum period (all, p<0.05)”

“The logistic regression analysis in this study showed a significant relationship between FH of psychiatric disorders and perinatal depressive symptomatology; thus, this control variable was included in the SEM models”.

  1. The title of Table 1 is misleading. I think the authors are showing two groups of women. The title makes me think that there is only one group of women with both antepartum and postpartum depression.

Thank you for pointing this out, we have now corrected this error and reported in the text “Tab. 1 Characteristics of women with either antepartum (N=234) or early postpartum depressive symptomatology (N=287)”

  1. It seems that a large number of women with depression have a family history of psychiatric problems (62.8 percent and 49.1 percent, respectively). This is a strong factor. I would suggest that the authors look at how the strategies for coping with stress work for these women in comparison to other women.

Please see the answer above.

“Furthermore, additional t-test analyses showed that significant differences in coping strategies between women with family history   and women without family history. Women with a family history of mental illness exhibited higher utilization of maladaptive avoidant coping strategies (i.e., behavioral disengagement, denial, self-distraction, and substance use) compared with women without family history during either pregnancy or post-partum period (all, p<0.05)”.

  1. It is also worthy to add some discussion about the relation between family history of psychiatric problems and depression.

We have added in the discussion section the following sentences:

“Finally, our results, in line with previous research, revealed a significant association between family history of any psychiatric illness and maternal peripartum depressive symptomatology [91] (https://pubmed.ncbi.nlm.nih.gov/35976654/). Additionally, we found a significative difference in maladaptive coping strategies between women with family history of mental illness compared with women without. These findings might suggest that risk for peripartum depressive symptomatology may be importantly influenced by the interaction between family history of mental illness and maladaptive coping strategies. However, our results, showed that coping strategies play a significant role in mediating the effect of attachment styles on depressive symptomatology even when controlling for the family history of psychiatric disorders. Further research is needed to explore the potential implications of the relationship between FH, coping and perinatal depressive symptomatology for developing targeted interventions and support strategies for women with family history of mental disorders.”

  1. What the numbers in Fig .1 and 2? It needs some description in the figure legend.

We have clarified this in the figures’ legend.

“ Figure 1 Path analysis model of the mediating role of coping in the relationship between attachment styles and antepartum depressive symptomatology.  Values represent standardized path coefficients, factor loadings and covariance estimates for the tested model. Solid lines rep-resent significant (p < .05), dashed lines non-significant effects (p < .05). The colored and bold lines represent significant pathways. Hum=Humour; Refr=Reframing; Acc=Acceptance; Plan=Planning; Act=Active coping; Sub = Substance Use; Den=Denial; Dis=Behavioral Disengagement; Bla=Sefl-Blame; Instr=Seeking Instrumental Support; Emot=Seeking Emotional Support; Vent=Venting of Emotions.”

Figure 2 Path analysis model of the mediating role of coping in the relationship between attachment styles and depressive symptomatology. Values represent standardized path coefficients, factor loadings and covariance estimates for the tested model. Solid lines rep-resent significant (p < .05), dashed lines non-significant effects (p < .05). The colored and bold lines represent significant pathways. Hum=Humour; Refr=Reframing; Acc=Acceptance; Plan=Planning; Act=Active coping; Den=Denial; Dis=Behavioral Disengagement; Bla=Sefl-Blame; Instr=Seeking Instrumental Support; Emot=Seeking Emotional Support; Vent=Venting of Emotions.

  1. Add some colors in the figures will make them easier to read.

Figure 1. Path analysis model of the mediating role of coping in the relationship between attachment styles and antepartum depressive symptomatology. Values represent standardized path coefficients, factor loadings and covariance estimates for the tested model. Solid lines represent significant (p < .05), dashed lines non-significant effects (p < .05). The colored and bold lines represent significant pathways. Hum=Humour; Refr=Reframing; Acc=Acceptance; Plan=Planning; Act=Active coping; Sub = Substance Use; Den=Denial; Dis=Behavioral Disengagement; Bla=Sefl-Blame; Instr=Seeking Instrumental Support; Emot=Seeking Emotional Support; Vent=Venting of Emotions

Figure 2. Path analysis model of the mediating role of coping in the relationship between attachment styles and postpartum depressive symptomatology.  Values represent standardized path coefficients, factor loadings and covariance estimates for the tested model. Solid lines represent significant (p < .05), dashed lines non-significant effects (p < .05).  The colored and bold lines represent significant pathways. Hum=Humour; Refr=Reframing; Acc=Acceptance; Plan=Planning; Act=Active coping; Den=Denial; Dis=Behavioral Disengagement; Bla=Sefl-Blame; Instr=Seeking Instrumental Support; Emot=Seeking Emotional Support; Vent=Venting of Emotions.

Reviewer 2 Report

Review: Coping as a mediator between attachment and depression either in pregnancy or in the early postpartum period: A structural equation modelling approach

This paper is about a relevant research and it brings a contribution to this field, pertaining to the mediating role of coping between attachment and perinatal depression. It is well structured and well written, and with appropriate methodology. I believe there are some features that could be clarified and improved, and some issues have implications for the Discussion.

Abstract:

The N of the sample should be presented.

Introduction:

p.1, line 36: “…including genetic predisposition, and various environmental, social, and psychological factors including negative personality traits” - What are negative personality traits? This sentence is not specific and lacks clarity, the authors should describe it.

Method:

P. 3, line 134: “The exclusion criteria included intellectual disability, unwillingness to participating in the study…” - It does not make sense to put unwillingness to participate as an exclusion criteria; these should be criteria defined by the researchers before the research takes place.

P. 4, line 189: The authors start describing global statistical analysis immediately after referring to the Coping questionnaire, in the same line: “Statistical analyses were performed”. - A subsection “Data analysis” should be at the end of the Method.  

Results:

P. 5, line 217: “…analyses leaving the antepartum total sample in 233.” - This is not coherent with the N in the Table title.

P.8, line 391: “Maladaptive coping was not associated with post-partum depressive symptoms” - Isn´t it antepartum?

P.9, line 403: “Moreover, results confirmed a mediating effect of maladaptive coping strategies on the relationships between 404 avoidant attachment and ante-partum depression” - Isn´t it emotional?

P.12, line 508: “In considering the relationship between attachment and coping the findings of the present study indicate a strong correlation between avoidant attachment style and emotional cop ing. As higher levels of attachment avoidance correspond to a lesser acknowledgment of distress and are related to a more discomfort in expressing difficulties to others [84], the observed positive correlation was unexpected.” At p.8, line 387, the authors have stated: “Avoidant attachment was associated with ante-partum depression, and less emotive coping strategies”. - This is incoherent, is the relation positive or negative? I believe it is negative. This must be clarified and of course it impacts the discussion.

P.12, line 556: “Even so, the questionnaire we used…”; - questionnaires

In the last paragraph, the authors state: “This would help a preventive coping-based intervention strategy to enhance the capacity of women with insecure attachment styles to use more adaptive coping  during and after pregnancy”. - at this point, it seems like both types of insecure attachment had the same results, so that they may be dealt with as the same. It was not the case, so a deeper and more specific implication could be explored.

Author Response

Review # 2: Coping as a mediator between attachment and depression either in pregnancy or in the early postpartum period: A structural equation modelling approach

This paper is about relevant research and it brings a contribution to this field, pertaining to the mediating role of coping between attachment and perinatal depression. It is well structured and well written, and with appropriate methodology. I believe there are some features that could be clarified and improved, and some issues have implications for the Discussion.

Abstract:

The N of the sample should be presented.

We have now added the N in the abstract. “. The aim of our study was to investigate the role of coping strategies in mediating the relationship between women’s attachment style and depressive symptomatology in pregnancy, and one week after giving birth, in a large cohort sample of women (N=1664)

Introduction:

p.1, line 36: “…including genetic predisposition, and various environmental, social, and psychological factors including negative personality traits” - What are negative personality traits? This sentence is not specific and lacks clarity, the authors should describe it.

We have now clarified this in the introduction “..psychological factors including negative personality traits (e.g., neuroticism, insecure attachments styles) and ineffective coping strategies (eg., denial, substance use).”

 Method:

  1. 3, line 134: “The exclusion criteria included intellectual disability, unwillingness to participating in the study…” - It does not make sense to put unwillingness to participate as an exclusion criteria; these should be criteria defined by the researchers before the research takes place.

We have now removed the inappropriate terms “unwillingness to participate in the study”.

  1. 4, line 189: The authors start describing global statistical analysis immediately after referring to the Coping questionnaire, in the same line: “Statistical analyses were performed”. - A subsection “Data analysis” should be at the end of the Method.

We have now created a subsection “Data analysis”.

 Results:

  1. 5, line 217: “…analyses leaving the antepartum total sample in 233.” - This is not coherent with the N in the Table title.

We have corrected the mistake: “antepartum total sample in 234”

 P.8, line 391: “Maladaptive coping was not associated with post-partum depressive symptoms” - Isn´t it antepartum?

We have corrected this error in the revised manuscript: “Emotional and maladaptive coping strategies were positively associated with antepartum depressive symptoms, while active coping strategies were negatively associated with antepartum depressive symptomatology.

P.9, line 403: “Moreover, results confirmed a mediating effect of maladaptive coping strategies on the relationships between 404 avoidant attachment and ante-partum depression” - Isn´t it emotional?

 “Moreover, results confirmed a mediating effect of emotional coping strategies on the relationships between avoidant attachment and antepartum depressive symptomatology”.

P.12, line 508: “In considering the relationship between attachment and coping the findings of the present study indicate a strong correlation between avoidant attachment style and emotional coping. As higher levels of attachment avoidance correspond to a lesser acknowledgment of distress and are related to a more discomfort in expressing difficulties to others [84], the observed positive correlation was unexpected.” At p.8, line 387, the authors have stated: “Avoidant attachment was associated with ante-partum depression, and less emotive coping strategies”. - This is incoherent, is the relation positive or negative? I believe it is negative. This must be clarified and of course it impacts the discussion.

We are very grateful to the reviewer for pointing out this potential source of confusion. We have now clarified this issue in this revised in the revised manuscript.

 “In considering the relationship between attachment and coping the findings of the present study indicate a negative correlation between avoidant attachment style and emotional coping.  As several emotional coping strategies (e.g., self-blame and venting of emotions) are intended to relieve the psychological distress without necessarily engaging with and relying on others for support, the observed negative correlation was unexpected [37]. However, those with attachment avoidance are inclined to a lesser recognition of distress and skeptical of trust in relationships. Thus, as previous research as demonstrated, the negative correlation between emotional coping and attachment avoidance could be attributable to the negative correlation between attachment avoidance and perceived stress as well as to the difficulty in expressing their feelings to others [84, 85].”

P.12, line 556: “Even so, the questionnaire we used…”; - questionnaires

We have now corrected this mistake.

 In the last paragraph, the authors state: “This would help a preventive coping-based intervention strategy to enhance the capacity of women with insecure attachment styles to use more adaptive coping  during and after pregnancy”. - at this point, it seems like both types of insecure attachment had the same results, so that they may be dealt with as the same. It was not the case, so a deeper and more specific implication could be explored.

We have now added in the conclusion the following phases: “Particularly our results suggest that different patterns of attachment and related coping strategies were associated with depressive symptoms during different phases of peripartum period. We found a significant indirect effect between attachment anxiety and maternal depressive symptoms via emotional and maladaptive coping strategies during pregnancy and via emotional coping strategies during the early postpartum period. Avoidant attachment style showed both a direct effect and an indirect effect through emotional coping on antenatal depressive symptoms, whereas no significant effects involving ineffective coping strategies on the pathway from attachment avoidance to depressive symptomatology were found during the postpartum period. The implications of these findings suggest that a better understanding of dynamics of women’s attachment and chosen coping strategies symptoms in different stages of peripartum period, may be useful in identifying and address risk factors associated with maternal depressive symptomatology. Further research would allow an exploration of the relationships between attachment and stress reactions as mediated by coping strategies. This would help a preventive coping-based intervention strategy to enhance the capacity of women with insecure attachment styles (especially anxious attachment) to use more adaptive coping during and after pregnancy.”.

Reviewer 3 Report

This study examines the relationship of coping strategies, attachment styles and symptoms of depression that occur in the late pregnancy period (third trimester) and within the early postpartum period (day 1–7). The investigated area is of great theoretical and applicative interest, but some methodological aspects adopted by the authors and some omitted considerations raise multiple doubts on the reliability and generalization of the results obtained.

EPDS extension

Some results collected with this research have been previously published in Frontiers in Psychiatry. In that manuscript, the authors adopted the same EPDS cut-off of ≥12 both in pregnancy and in the postpartum period.

Although it is the same study sample, why in the manuscript proposed to Brain Sciences do the authors adopt different EPDS cut-offs for pregnancy (≥12) and postpartum (≥9)?

In this regard, it should be noted that there is currently no Italian validation of the EPDS for its use in pregnancy. Therefore, the choice of the EPDS cut-off ≥12 is subjective and refers to validations of the instrument involving non-Italian pregnant women.

Despite the multiple developments and the diffusion of the EPDS all over the world, some critical issues have recently been reported regarding the use of the instrument during the first postpartum week [see Lee et coll., Psychosom Med, 2003;65(3):357-61; Ezirim et coll., Am J Perinatol. 2023;40(2):194-200]. In terms of detecting women at risk of postnatal depression, is it appropriate on the 3rd/4th postpartum day to administer a tool such as the EPDS and ask a woman how she has been feeling in the past 7 days?

Finally, despite the clear indications provided by Cox and colleagues, the authors of the manuscript adopt the EPDS cut-off in dichotomous terms, consequently placing the sample in two sub-samples with absence/presence of depression. This method is improper and misleading.

Instead, exceeding the cut-off simply enables identification of women with high depression symptoms. Therefore, the wording adopted by the authors on perinatal depressive symptomatology should be completely revised throughout the manuscript, including the title.

DATE

The data of this study were collected in June-November 2020 during the first severe wave of the COVID-19 pandemic.

In almost all Italian hospitals, women in labor could not be accompanied by their partner, peripartum medical visits were limited as was contact with people and health professionals. Stress, loneliness, fear of contagion and multiple restrictions characterized the perinatal experience in pregnant and postpartum women.

Therefore, the exclusive pre-COVID-19 bibliographic references used by the authors omit a clinical condition that characterized the study sample. With this choice, the authors therefore seem not to fully grasp the meaning of the results obtained. In this historical moment it would in fact be more appropriate to give an interpretation of the results obtained as well as to draw from the abundant literature on perinatal mental health and COVID-19.

Author Response

Reviewer #3.

This study examines the relationship of coping strategies, attachment styles and symptoms of depression that occur in the late pregnancy period (third trimester) and within the early postpartum period (day 1–7). The investigated area is of great theoretical and applicative interest, but some methodological aspects adopted by the authors and some omitted considerations raise multiple doubts on the reliability and generalization of the results obtained.

EPDS extension

Some results collected with this research have been previously published in Frontiers in Psychiatry. In that manuscript, the authors adopted the same EPDS cut-off of ≥12 both in pregnancy and in the postpartum period. Although it is the same study sample, why in the manuscript proposed to Brain Sciences do the authors adopt different EPDS cut-offs for pregnancy (≥12) and postpartum (≥9)?

The aim of our study was to investigate the relationship between attachment styles and perinatal depression through the mediating role of coping strategies. Many previous studies investigating the relationship between insecure attachment styles and postpartum depression reported a significant association between insecure attachment styles and postpartum depression symptomatology using EPDS with a cutoff score of  > 9/10 ( https://pubmed.ncbi.nlm.nih.gov/25562537/ ; https://pubmed.ncbi.nlm.nih.gov/23426862/; https://pubmed.ncbi.nlm.nih.gov/25098625/ ).

The recommended EPDS cut-off score of 9 or more was also used to investigate the relationship between personality features such as neuroticism and  clinical levels of depression https://www.ncbi.nlm.nih.gov/pmc/articles/PMC3359638/ ; https://pubmed.ncbi.nlm.nih.gov/25536977/).

Several studies suggested that the optimal EPDS cut-off scores change considerably during different stages of peripartum period and many authors recommended  an optimal EPDS cut off score >9 for identifying patients with major depression, during the postpartum period https://pubmed.ncbi.nlm.nih.gov/15458557/ ; https://pubmed.ncbi.nlm.nih.gov/17068676/; https://pubmed.ncbi.nlm.nih.gov/35764977/ https://onlinelibrary.wiley.com/doi/10.1111/pcn.12562; https://pubmed.ncbi.nlm.nih.gov/32584147/ This cut-off point has been recommended for identifying depressive symptomatology in the immediate postnatal period  https://pubmed.ncbi.nlm.nih.gov/16986815/  https://pubmed.ncbi.nlm.nih.gov/15458557/ and has been shown to have high sensitivity, specificity, and positive predictive power for postpartum depression https://pubmed.ncbi.nlm.nih.gov/2224383/

Therefore, in agreement with these previous studies, we used the cut-off EPDS score ≥ 9 as a measure for postpartum depression.

In this regard, it should be noted that there is currently no Italian validation of the EPDS for its use in pregnancy. Therefore, the choice of the EPDS cut-off ≥12 is subjective and refers to validations of the instrument involving non-Italian pregnant women.

However, we have now included this statement in the limitations section:

We agree with the reviewer. Two studies have validated the EPDS postnatally in Italy. Carpiniello et al. 1997 https://pubmed.ncbi.nlm.nih.gov/9443138/ reported optimal screen-positive cut-off scores for depressive symptoms of 9 or more, while Benvenuti et al. 1999 (https://pubmed.ncbi.nlm.nih.gov/10360408) reported that a score of 12 or more detected all six women in their study with major depression. However, there has not yet been an Italian validation study for the of EPDS in pregnancy.

We have now added these sentences in the limitations section:

“Although there has not yet been an Italian validation study for the use of EPDS in pregnancy, we used EPDS to assess antenatal depressive symptomatology. However, according to previous Italian studies which investigated the prevalence and determinants of antenatal depression among Italian mothers, we used a cut-off > 12 [54, 93] https://pubmed.ncbi.nlm.nih.gov/33069119/ ; https://pubmed.ncbi.nlm.nih.gov/22526826/ )”.

Despite the multiple developments and the diffusion of the EPDS all over the world, some critical issues have recently been reported regarding the use of the instrument during the first postpartum week [see Lee et coll., Psychosom Med, 2003;65(3):357-61; Ezirim et coll., Am J Perinatol. 2023;40(2):194-200]. In terms of detecting women at risk of postnatal depression, is it appropriate on the 3rd/4th postpartum day to administer a tool such as the EPDS and ask a woman how she has been feeling in the past 7 days?

We agree with the reviewer. This is an important issue that requires more research to resolve. While the articles cited reported that the EPDS did not predict the development of postpartum depression, at the 6-week postpartum time interval, when administered in the immediate postpartum period, it should be noted that the immediate postpartum period was defined as between 3 and 24 hours postpartum by Ezirim et coll., Am J Perinatol. 2023;40(2):194-200 and within 48 hours of delivery by Lee et coll., Psychosom Med, 2003;65(3):357-61, a period generally regarded as unlikely for postnatal depressive symptomatology.

Numerous other recent studies have validated the EPDS as a screening instrument to detect women at risk of postpartum depression within one week postpartum and demonstrated that the EPDS has good sensitivity, specificity and predictive power in the immediate postpartum period to identify mothers at risk for postpartum depression symptomatology at 4 and 8 weeks postpartum (see above). We built on these results in deciding to use the EPDS as early as one week after delivery.

Finally, despite the clear indications provided by Cox and colleagues, the authors of the manuscript adopt the EPDS cut-off in dichotomous terms, consequently placing the sample in two sub-samples with absence/presence of depression. This method is improper and misleading.

Instead, exceeding the cut-off simply enables identification of women with high depression symptoms. Therefore, the wording adopted by the authors on perinatal depressive symptomatology should be completely revised throughout the manuscript, including the title.

We have now revised the manuscript according to the reviewer’s suggestions.

DATE

The data of this study were collected in June-November 2020 during the first severe wave of the COVID-19 pandemic.

In almost all Italian hospitals, women in labor could not be accompanied by their partner, peripartum medical visits were limited as was contact with people and health professionals. Stress, loneliness, fear of contagion and multiple restrictions characterized the perinatal experience in pregnant and postpartum women.

Therefore, the exclusive pre-COVID-19 bibliographic references used by the authors omit a clinical condition that characterized the study sample. With this choice, the authors therefore seem not to fully grasp the meaning of the results obtained. In this historical moment it would in fact be more appropriate to give an interpretation of the results obtained as well as to draw from the abundant literature on perinatal mental health and COVID-19.

We agree that this is an important point which was discussed in the original paper (Bellomo et al., 2002) We added these sentences in the Discussion section.

“Our data collection was made in the first months of the first wave of the COVID-19 pandemic. Several previous studies specifically exploring the impact of COVID-19 on maternity, during the first severe wave of the COVID-19 pandemic (2020), reported a significant higher prevalence and levels of depressive symptomatology in Italian mothers during the pandemic than previously reported in literature (; https://www.ncbi.nlm.nih.gov/pmc/articles/PMC7704433/  https://pubmed.ncbi.nlm.nih.gov/33208115/

https://pubmed.ncbi.nlm.nih.gov/32751804/  ). However, results have been inconsistent and other findings did not show significant differences in depressive and anxiety symptoms in women who delivered after the implementation of COVID-19 restrictions and threat from pre-pandemic women, suggesting that reproductive experience alters the female brain in adaptive ways. https://www.ncbi.nlm.nih.gov/pmc/articles/PMC9866377/ ; https://pubmed.ncbi.nlm.nih.gov/32751804/  We cannot rule out the possibility that our findings have been influenced by high levels of perceived stress among pregnant women during the COVID-19. Unfortunately, we did not measure pandemic-related stress symptoms. However, it should be noted that pregnant women are likely to use avoidant coping strategies like substance abuse and self-distraction for dealing with the stressful COVID-related situation (https://www.ncbi.nlm.nih.gov/pmc/articles/PMC7704433/ ). In our sample, instead, women with insecure attachment styles choose more frequently emotional coping strategies. More research is needed to clarify the impact of COVID 19 on coping strategies and depressive symptomatology during and after pregnancy”.

Round 2

Reviewer 3 Report

The authors intervened massively on the contents of the manuscript, but some further additions and corrections are necessary.

In particular, it should be noted throughout the text and in the tables of the manuscript that the associations described refer to subjects with high depressive symptomatology.

2.1. Edinburgh Postnatal Depression Scale.

(page 4, lines 170-174): The EPDS was initially configured by Cox et coll. (1987) to detect the level of maternal depressive symptoms present in the first weeks after childbirth and, consequently, to identify women at psychopathological risk through a clinical interview. Only later was the EPDS validated for use in the perinatal period.

(lines 182-183): Many “international” studies …..

2.2. Experiences in Close relationship Scale.

Cite some Italian studies that in the perinatal period - if any - have used this tool.

2.3. Coping Orientation for Problem Experiences

Cite some Italian studies that in the perinatal period - if any - have used this tool.

LIMITATIONS OF THE STUDY

It should be noted that the EPDS detects only the symptomatological level and is therefore not a diagnostic tool. In fact, women with high depressive symptoms could have bipolar disorder, mood disorder, adjustment disorder or other.

Therefore, the significant associations with high level of depressive symptoms identified by the authors should be viewed with caution and prudence.

Author Response

We are very grateful to the reviewer for her/his constructive suggestions.

2.1. Edinburgh Postnatal Depression Scale.

(page 4, lines 170-174): “The EPDS was initially configured by Cox et coll. (1987) to detect the level of maternal depressive symptoms present in the first weeks after childbirth and, consequently, to identify women at psychopathological risk through a clinical interview. Only later was the EPDS validated for use in the perinatal period “.

(lines 182-183): Many “international” studies …..

The manuscript has been now revised accordingly.

 “The EPDS was originally configured by Cox et coll. [51] to detect the level of maternal depressive symptoms present in the first weeks after childbirth and, consequently, to identify women at psychopathological risk through a clinical interview.  Subsequent studies have validated the EPDS for use in the perinatal period (during pregnancy and postpartum) [52, 53]”

2.2. Experiences in Close relationship Scale.

Cite some Italian studies that in the perinatal period - if any - have used this tool.

“We employed the Italian version of the ECR [64], which was used in previous Italian studies on the relationship between attachment patterns and perinatal depression [65, 66].” https://pubmed.ncbi.nlm.nih.gov/26798510/         https://pubmed.ncbi.nlm.nih.gov/26156819/

2.3. Coping Orientation for Problem Experiences

Cite some Italian studies that in the perinatal period - if any - have used this tool.

“This scale was previously used in a sample of Italian mothers to study the link between coping and postpartum depressive symptoms [70]”.  https://pubmed.ncbi.nlm.nih.gov/33312140/

LIMITATIONS OF THE STUDY

It should be noted that the EPDS detects only the symptomatological level and is therefore not a diagnostic tool. In fact, women with high depressive symptoms could have bipolar disorder, mood disorder, adjustment disorder or other.

Therefore, the significant associations with high level of depressive symptoms identified by the authors should be viewed with caution and prudence.

 We have added the following sentence in the Limitation section:

“Another limitation of the study is the absence of a clinical interview; thus, we could not directly compare

EPDS scores with the clinical diagnosis of major depression”.
